# PeerJ

# Approaches to describing inter-rater reliability of the overall clinical appearance of febrile infants and toddlers in the emergency department

Paul Walsh[1], Justin Thornton[2], Julie Asato[2], Nicholas Walker[2], Gary McCoy[2], Joe Baal[2], Jed Baal[2], Nanse Mendoza[2] and Faried Banimahd[3]

[1] Department of Emergency Medicine, University of California Davis Medical Center, Sacramento, CA, USA
[2] Department of Emergency Medicine, Kern Medical Center, Bakersfield, CA, USA
[3] Department of Emergency Medicine, University of California Irvine, Orange, CA, USA

Corresponding author
Paul Walsh, pfwalsh@ucdavis.edu

## ABSTRACT

**Objectives.** To measure inter-rater agreement of overall clinical appearance of febrile children aged less than 24 months and to compare methods for doing so.

**Study Design and Setting.** We performed an observational study of inter-rater reliability of the assessment of febrile children in a county hospital emergency department serving a mixed urban and rural population. Two emergency medicine healthcare providers independently evaluated the overall clinical appearance of children less than 24 months of age who had presented for fever. They recorded the initial 'gestalt' assessment of whether or not the child was ill appearing or if they were unsure. They then repeated this assessment after examining the child. Each rater was blinded to the other's assessment. Our primary analysis was graphical. We also calculated Cohen's $\kappa$, Gwet's agreement coefficient and other measures of agreement and weighted variants of these. We examined the effect of time between exams and patient and provider characteristics on inter-rater agreement.

**Results.** We analyzed 159 of the 173 patients enrolled. Median age was 9.5 months (lower and upper quartiles 4.9–14.6), 99/159 (62%) were boys and 22/159 (14%) were admitted. Overall 118/159 (74%) and 119/159 (75%) were classified as well appearing on initial 'gestalt' impression by both examiners. Summary statistics varied from 0.223 for weighted $\kappa$ to 0.635 for Gwet's AC2. Inter rater agreement was affected by the time interval between the evaluations and the age of the child but not by the experience levels of the rater pairs. Classifications of 'not ill appearing' were more reliable than others.

**Conclusion.** The inter-rater reliability of emergency providers' assessment of overall clinical appearance was adequate when described graphically and by Gwet's AC. Different summary statistics yield different results for the same dataset.

## INTRODUCTION

Deciding whether a febrile child is 'ill appearing' is a key decision point in emergency department (ED) management algorithms for febrile infants and toddlers (*Baker, Bell & Avner, 1993*; *Baraff et al., 1993*; *Baskin, O'Rourke & Fleisher, 1992*; *Jaskiewicz et al., 1994*). Initial physician judgments of this overall appearance are generally made rapidly and prior to completing a full physical examination. Such judgments can even affect how providers interpret clinical findings (*McCarthy et al., 1985*).

Implicit in this construct is the assumption that clinicians can agree on whether or not a child is ill appearing. There is little evidence that addresses the inter-rater reliability of providers' overall impression of febrile children's appearance. One study found good agreement for individual clinical signs many of which are associated with overall clinical appearance and often with fever (*Wagai et al., 2009*). Others have addressed inter-rater reliability for the Yale observation score (*McCarthy et al., 1985*); but studies of overall clinical impression without the use of specific scoring systems are scarce. The inter-rater reliability of individual historical and examination findings has been studied for a variety of conditions including diagnostic interviews, head trauma and bronchiolitis (*Holmes et al., 2005*; *Shaffer et al., 1993*; *Walsh et al., 2006*). Establishing adequate inter-rater reliability is an important component in the derivation of clinical management algorithms (*Laupacis, Sekar & Stiell, 1997*; *Stiell & Wells, 1999*) but is often not performed (*Maguire et al., 2011*).

Although clinical appearance is a binary decision node in management algorithms (*Baker, Bell & Avner, 1993*; *Baraff et al., 1993*; *Jaskiewicz et al., 1994*), clinical appearance is a continuum as some children appear more ill than others. When given the option providers chose 'unsure' in 12.6% of infants and toddlers presenting to an ED in one study (*Walsh et al., 2014*). These children in whom the provider was "unsure" had historical and physical examination findings intermediate in severity between those classified as 'ill' and 'not ill' appearing. The prevalence of bacterial meningitis and pneumonia was also intermediate between those classified as ill or not ill appearing (*Walsh et al., 2014*).

Despite the widespread use of management strategies that rely on overall clinical appearance, the inter-rater reliability of clinical appearance is not well established. Moreover, because ill appearing children are in a small minority, widely used measures of inter-rater reliability such as Cohen's $\kappa$ statistic risk being overly conservative. This is also true for other summary measures of inter rater agreement that rely on the marginal distribution of categories. Consequently even though actual agreement (reliability) between raters is high the summary statistic will be low. In the context of clinical decision research this could lead to useful clinical characteristics being incorrectly labeled too unreliable for clinical use. Alternative approaches, including simple graphical analysis, are not widely used in medicine.

The first aim of this study was to measure inter-rater agreement of overall clinical appearance of febrile children aged less than 24 months. We hypothesized that inter-rater agreement of overall clinical appearance would be adequate for clinical use. In addition, we hypothesized that agreement is influenced by the clinical experience of raters. The

second aim of this study was to compare methods for evaluating inter-rater agreement in unbalanced samples and in particular examine graphical methods.

## METHODS

### Design

We conducted a cross sectional observational study of inter-rater reliability performed in accordance with the guidelines for reporting reliability and agreement studies (*Kottner et al., 2011*). The study was approved by Kern Medical Center institutional review board (IRB) (approval #10032). Our IRB did not require written consent from our healthcare providers.

### Setting

The study was performed at a county hospital teaching emergency department (ED) with emergency medicine residency.

### Subjects

The subjects were 9 board-eligible or certified general emergency medicine physicians, a pediatric emergency physician, three mid-level providers and 21 residents for a total of 34 providers. The patients in the study were children aged less than 24 months who presented to the ED with a chief complaint of fever, or a rectal temperature of at least 38 °C at triage.

### Implementation

Eligible patients were identified by research assistants (RA) or physician investigators. RA coverage was typically available 12–16 h a day, seven days a week including at night and holidays. We enrolled a convenience sample between February 2012 and April 2013 inclusive. Enrollment could be performed only when trained research staff or investigators were immediately available. RAs identified potentially eligible patients based on their presenting complaint recorded at triage. The RA obtained verbal consent from the parents and provided the parents with informational brochure describing the study in lieu of written consent. After this the RA waited with the patient until the provider arrived to see the patient.

This provider was identified as the first rater. The provider was asked then handed his/her datasheet and asked provide their immediate 'gestalt' assessment of whether they felt the child was 'ill' or 'not ill' appearing or if the provider was 'unsure'. The provider was then asked to complete their physical examination and again answer whether they felt the child was 'ill' or 'not ill' appearing or if the provider was 'unsure'. The provider was not allowed to change their original assessment following the examination. The examination to be performed was not prescribed, rather, physicians were asked to complete the physical examination they would normally do for this child.

The RA then identified a second provider. When doing this the RAs were instructed to approach the first available other provider. This was to prevent bias that could occur if RAs were allowed to choose providers who they found more approachable. The process was

repeated for the second provider. Neither provider was allowed see the other's data sheet or discuss the case prior to the second evaluation.

The RAs were instructed to encourage the second provider perform his or her evaluation within 10 min of the first. Because this would often not be possible we recorded the time each evaluation was started for later sensitivity analysis. We did not record the amount of time each provider took to complete their evaluation to allay possible provider concerns that such information might be re-used in other ways. We also recorded if antipyretics were given prior to the first rater's evaluation or between the first and second raters' evaluations.

The level of training of each provider was recorded by the RA. The final disposition and diagnosis was recorded by the RA at the end of the patient's ED visit. The data were recorded on two data forms, one for each provider. RAs timed the data sheets and entered the results into a customized database (Filemaker Pro, Santa Clara, CA).

We performed an initial pilot implementation. Based on this pilot experience the providers were provided with the following guidance: 'If you feel a child is 'ill appearing' some intervention, for example: laboratory or radiology testing, or reevaluation following antipyretics and observation, should occur prior to discharge from the ED. If you are uncertain whether the child is either 'ill' or 'not ill appearing' then 'unsure' is the appropriate category'. No more specific guidance than this was provided. The pilot phase also led to strict research assistant (RA) procedures to ensure that the two providers could not confer on a case until after both data collection forms had been seized. Data from the pilot implementation period were not included in the analysis.

## Outcomes

Our primary clinical outcome was the inter-rater agreement between of the immediate 'gestalt' impression of the clinical appearance of the two providers. Our secondary outcomes were:

(1) Inter-rater agreement between the providers' assessment of clinical appearance after examining the patient.

(2) Intra-rater agreement between providers' immediate 'gestalt' impression and their impression after the exam. Poor intra rater reliability between providers' own gestalt assessment and their assessment after examining the child would invalidate gestalt assessments of clinical appearance.

We performed sensitivity analysis examining the effect of time between the two assessments. We used logistic regression to analyze the effect of antipyretics in triage and between evaluations, the child's age, the experience of the raters, the interval between first and second evaluations. We also sought interactions between the interval between evaluations, age and antipyretic use. Variables were selected based on biological plausibility rather than using automated stepwise techniques. Variables were retained in the final model for $p \leq 0.05$.

## Rationale for statistical methods

Simple percentage agreement may be an adequate measure of agreement for many purposes, but does not account for agreement arising from chance alone. Attempts to account for the agreement that may arise from chance have led to a variety of methods for measuring inter-rater reliability. These methods vary with different approaches for continuous, ordinal, categorical, and nominal data (*Cohen, 1960*; *Fleiss, 1971*; *Cohen, 1968*; *Banerjee et al., 1999*). Our data could be considered categorical but, based on the ordinal association between components of clinical appearance and some microbiological outcomes, our classification scheme could also be considered ordinal. We used both approaches.

Categorical agreement is often measured with Cohen's $\kappa$. Cohen's $\kappa$ appears easily interpreted; its minimum and maximum are $-1$ and $+1$ for $2 \times 2$ tables. For a $k \times k$ table the minimum is $-1/(k-1)$ and approaches 0 as $k$ gets larger; while the maximum is always $+1$. Negative values imply disagreement beyond independence of raters and positive values agreement beyond independence of raters. Descriptive terms such as 'moderate' and 'poor' agreement have been published to further ease interpretation (*Landis & Koch, 1977*). The simple $\kappa$ assumes two unique raters.

When the two raters' identities vary an implementation of the more than two raters case must be used (*Fleiss, Levin & Paik, 2003*; *Statacorp, 2013*). A provider could be the first reviewer for one infant and the second reviewer for another. We did this because our question was about the inter-rater reliability of providers in general rather than any specific provider pair. Consequently the study we carried out was one of many that could have been carried out; by reversing the order of which provider was selected as reviewer 1 and reviewer 2 one could conceivably obtain different $\kappa$ scores even though the percentage agreement would be unchanged. Assuming there was no bias in how we selected first and second reviewers we anticipated this effect would be small given the kappa calculation we used. We simulated 500 alternative permutations of the order of reviewers to verify this assumption.

The best design for a $k \times k$ table to measure agreement is one where the margins have roughly a proportion of $1/k$ of the total sample studied; in the $2 \times 2$ case this means a prevalence of 0.5 for the indication as well as its compliment. Serious deviations from this are known to make the variance for $\kappa$ unstable and $\kappa$ misleading for measuring agreement amongst the $k$ levels of the scale. However such evenly distributed margins are unrepresentative of most clinical scenarios, particularly in pediatric emergency medicine where non-serious outcomes often far outnumber serious ones. A disadvantage of the $\kappa$ statistic is that it results in lower values the further the prevalence of the outcome being studied deviates from 0.5 (*Feinstein & Cicchetti, 1990*; *Gwet, 2008*). Scott's $\pi$ (subsequently extended by Fleiss) suffers the same limitations (*Scott, 1955*). This so called '$\kappa$ paradox' is well described and understood by statisticians. When interpreted by others, however, this property of $\kappa$ could lead to clinical tools with potentially useful but imperfect reliability being discarded based on a low reported $\kappa$ value. Consequently $\kappa$ and Scott's $\pi$ risk misinterpretation when one of the categories being rated is much more

or less common than the other. Gwet developed an alternative method, the agreement coefficient ($AC_1$) specifically to address this limitation (*Gwet, 2008*). The $AC_1$ has potential minimum and maximum values of $-1$ and $+1$ respectively. The $AC_1$ is more stable than $\kappa$ although the $AC_1$ may give slightly lower estimates than $\kappa$ when the prevalence of a classification approaches 0.5 but gives higher values otherwise (*Gwet, 2008*). The $AC_1$ does not appear widely in the medical literature despite recommendations to use it (*McCray, 2013*; *Wongpakaran et al., 2013*). This may be because of two key assumptions; (1) that chance agreement occurs when at least one rater rates at least some individuals randomly and (2) that the portion of the observed ratings subject to randomness is unknown. On the other hand these assumptions may not be stronger than those inherent in Cohen's $\kappa$.

Ordinal agreement can be measured using a weighted $\kappa$. The penalty for disagreement is weighted according to the number of categories by which the raters disagree (*Cohen, 1968*). The results are dependent both on the weighting scheme chosen by the analyst and the relative prevalence of the categories (*Gwet, 2008*). One commonly recommended weighting scheme reduces the weighted $\kappa$ to an intra-class correlation (*Fleiss & Cohen, 1973*). Scott's $\pi$ and Gwet's $AC_1$ can also be weighted. When weighted, Gwet's $AC_1$ is referred to as $AC_2$ (*Gwet, 2012*).

Another approach is to regard ordinal categories as bins on a continuous scale. Polychoric correlation estimates the correlation between raters as if they were rating on a continuous scale (*Flora & Curran, 2004*; *Uebersax, 2006*). Polychoric correlation is at least in principle insensitive to the number of categories and can even be used where raters use different numbers of categories. The correlation coefficient, $-1$ to $+1$, is interpreted in the usual manner. A disadvantage of polychoric correlation is that it is susceptible to distribution; although some recognize polychoric correlation as a special case of latent trait modeling thereby allowing relaxation of distribution assumptions (*Uebersax, 2006*). The arguments against using simple correlation as a measure for agreement for continuous variables in particular have been well described (*Bland & Altman, 1986*).

It is easy to conceive that well appearing infants are more common than ill appearing ones, thereby raising concerns that assumptions of a normal distribution are unlikely to hold. Another coefficient of agreement "A" proposed by van der Eijk was specifically designed for ordinal scales with a relatively small number of categories dealing with abstract concepts. This measure "A" is insensitive to standard deviation. "A" however contemplates large numbers of raters rating a small number of subjects (such as voters rating political parties) (*Van der Eijk, 2001*).

The decision to use of a single summary statistic to describe agreement is fraught with the risks of imbalance in the categories being rated, different results from different methods and the need to ordain in advance a specific threshold below which the characteristic being classified will be discarded as too unreliable to be useful for decision making.

We used a simple graphical method for our primary analysis. For the graphical method we categorized agreement as follows:

| | |
|---|---|
| Reviewer 1 and reviewer 2 agree | Ill appearing: ill appearing<br>Not ill appearing: not ill appearing<br>Unsure: unsure |
| Reviewer 1 considers infant more ill appearing by one category than reviewer 2 | Ill appearing: unsure Unsure: Not ill appearing |
| Reviewer 1 considers patient more ill appearing by two categories than reviewer 2 | Ill appearing: Not ill appearing |
| Reviewer 1 considers patient less ill appearing by one category than reviewer 2 | Unsure: ill appearing<br>Not ill appearing : unsure |
| Reviewer 1 considers patient less ill appearing by two categories than reviewer 2 | Not ill appearing: Ill appearing |

We created a bar graph with a bar representing the percentage of patients in each category and, by simulation in an exemplary dataset, a graph to portray how random assignment of categories would appear. This graph would be expected to be symmetrical around the bar portraying when the providers agreed. Asymmetry could suggest bias or suggest or that a change in the quantity being measured has occurred between the two exams. This could arise if the infants' condition changed between the two exams. We created an artificial dataset where agreement was uniformly randomly assigned and used this to create a reference graph of what random agreement alone would look like. All graphs were drawn using Stata 13.

We also calculated weighted kappa ($\kappa$) using widely used weighting schemes, polychoric correlation, and Gwet's agreement coefficient ($AC_1$ and $AC_2$) as secondary methods (*Gwet, 2008*).

We performed sensitivity analysis using logistic regression to examine the effect age, diagnosis, antipyretic use, experience levels of the raters, and time between evaluations. We analyzed the rater pairs as using several strategies. In one we assigned an interval value for each year of post graduate training with attending physicians all assigned a value of six. In another strategy we grouped residents as Post graduate year (PGY)1 and PGY2, PGY3 and PGY4, Mid level provider (MLP) and attending, assigned values of 1–4 and analyzed these. We also examined rater pair combinations as nominal variables.

## Sample size calculations

Given the lack of sample size methods for graphical analysis we relied on sample size calculations for a traditional Cohen's $\kappa$. We assumed that 75% of each raters' classifications would be for the more common outcome, a $\kappa$ of 0.8, an absolute difference range in the final $\kappa$ of $\pm0.1$ and an $\alpha$ of 0.05 (*Reichenheim, 2000*). This resulted in a sample size of 144 patients.

Data management, logistic, $\kappa$ and polychoric (*Kolenikov, 2004*) estimations were performed using Stata version 13.0 software (Statacorp LLP, College Station, TX). Gwet's AC1 was calculated using R Version 3.01, (www.r-project.org. AC1 function from *Emmanuel, 2013*). Other measures of agreement were estimated using Agreestat, (www.agreestat.com).

**Table 1  Diagnoses by gestalt classification for each rater.** Percentages exceed 100 because of rounding.

| Diagnosis | N (%) | First rater 'gestalt' impression | | | Second rater 'gestalt' impression | | |
|---|---|---|---|---|---|---|---|
| | | Not ill | Unsure | Ill | Not ill | Unsure | Ill |
| Pneumonia | 14 (9) | 3 | 3 | 8 | 3 | 1 | 10 |
| UTI or Pyelonephritis | 4 (3) | 1 | 0 | 3 | 1 | 0 | 3 |
| Bronchiolitis | 9 (6) | 2 | 0 | 7 | 2 | 1 | 6 |
| Otitis media | 6 (4) | 1 | 1 | 4 | 1 | 1 | 4 |
| Gastroenteritis | 8 (5) | 1 | 1 | 6 | 0 | 0 | 8 |
| Cellulitis | 6 (4) | 4 | 0 | 2 | 3 | 0 | 3 |
| Sepsis No focus | 3 (2) | 1 | 0 | 2 | 1 | 0 | 2 |
| URI | 36 (23) | 3 | 3 | 30 | 5 | 1 | 30 |
| Herpangina | 2 (1) | 0 | 0 | 2 | 0 | 0 | 2 |
| Pharyngitis | 2 (1) | 0 | 0 | 2 | 0 | 0 | 2 |
| Viral/Febrile illness NOS | 46 (29) | 5 | 5 | 36 | 4 | 5 | 37 |
| Bacteremia | 1 (1) | 0 | 0 | 1 | 0 | 0 | 1 |
| Varicella | 2 (1) | 0 | 1 | 1 | 0 | 0 | 2 |
| Febrile Seizure | 5 (3) | 2 | 0 | 3 | 2 | 0 | 3 |
| Non infective | 3 (2) | 0 | 0 | 3 | 0 | 0 | 3 |
| Other Febrile illness | 12 (8) | 4 | 0 | 8 | 5 | 0 | 7 |
| Total | 159 | 27 | 14 | 118 | 27 | 9 | 123 |

**Notes.**

UTI, Urinary tract infection; URI, Upper respiratory tract infection; NOS, not otherwise specified.

## RESULTS

We analyzed 159 of the 173 patients enrolled. We excluded 14 visits, 12 for age violations, and two for repeated enrollments (only the index visit was retained). There were 99/159 (62%) boys and the median age was 9.5 months, (lower and upper quartiles 4.9, 14.6 months) and 22/159 (14%) were admitted. Eighty (50%) patients received antipyretics prior to evaluation by both providers, and 25/159 (16%) received antipyretics between the first and second provider's assessments. The ED diagnoses are summarized in Table 1.

### Combinations of raters

We observed 29 different combinations in the order of evaluations and level of provider training. These are described in Table 2. The minimum number of evaluations performed by any individual provider was 1/358 (0.3%) the maximum was 16/358 (4%).

### Time between evaluations

The median time between the start of the first and second raters' evaluations was 20 min (interquartile range 6–45 min) and is shown in Fig. 1. Antipyretics were given between the evaluations in 24 cases.

### Agreement

Overall 118/159 (74%) and 119/159 (75%) were classified as well appearing on initial 'gestalt' impression by the two examiners. When the first rater classified a child as well appearing the second was more likely to agree 94/120 (78%) than when the first rater

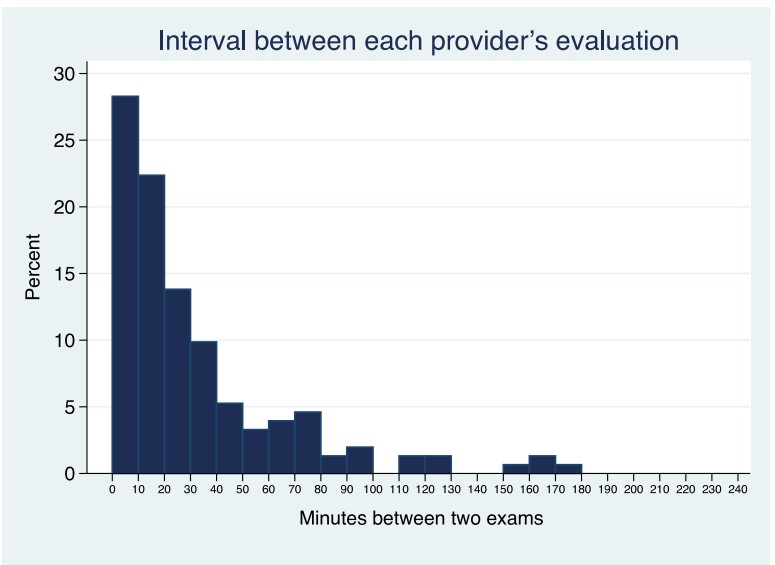

**Figure 1  Time between evaluations.** Histogram showing the time interval between the first and second raters' gestalt assessment of clincial appearance. Six outliers with interval >240 min are not shown.

**Table 2  The levels of training of the first and second raters and numbers of patients seen by each combination.**

| First rater | Second rater | | | | |
|---|---|---|---|---|---|
| | PGY1/2 | PGY3/4 | MLP | Attending | Total |
| **PGY1/2** | 6 (4) | 16 (10) | 2 (1) | 6 (4) | 30 (19) |
| **PGY3/4** | 9 (6) | 33 (21) | 3 (2) | 29 (18) | 74 (47) |
| **MLP** | 5 (3) | 21 (13) | 8 (5) | 6 (4) | 40 (25) |
| **Attending** | 4 (3) | 11 (7) | 0 (0) | 0 (0) | 15 (9) |
| **Total** | 24 (15) | 81 (51) | 13 (8) | 41 (26) | 159 (100) |

Notes.
PGY, Post graduate year of training; MLP, Mid-level provider; Attending, Board certified emergency physician.
Rounded percentages in parentheses.

classified the child as ill appearing 8/27 (30%) $p = 0.025$. The agreement between all raters and categories are shown in Tables 3 and 4; intra rater agreement is shown in Table 5.

The weighted $\kappa$ was 0.223 for initial gestalt assessment and 0.314 following examination. Our simulation comparing 500 random alternative permutations of first and second reviewers found little evidence of bias. Where our observed weighted $\kappa$ was 0.223 the minimum and maximum found in our alternative possible reviewer order permutations was 0.220–0.235. This argues against bias resulting from the order in which the raters were selected.

The polychoric correlation for initial 'gestalt' assessment and assessment following examination were 0.334 and 0.482 respectively. These, the weighted $\kappa$, and the AC$_2$ all point to increased agreement after the examiners had completed a full exam of the infant.

**Table 3** Raw inter-rater agreement for immediate 'gestalt' impression of the child's overall clinical appearance.

| First rater gestalt impression | Second rater gestalt impression | | | Total |
|---|---|---|---|---|
| | Not ill appearing | Unsure | Ill appearing | |
| Not ill appearing | 94 | 11 | 13 | 118 |
| Unsure | 12 | 0 | 2 | 14 |
| Ill appearing | 14 | 5 | 8 | 27 |
| **Total** | 120 | 16 | 23 | 159 |

**Table 4** Raw inter-rater agreement for clinical impression of the child's overall appearance after examining the child.

| First rater after examining child | Second rater after examining child | | | Total |
|---|---|---|---|---|
| | Not ill appearing | Unsure | Ill appearing | |
| Not ill appearing | 103 | 6 | 14 | 123 |
| Unsure | 8 | 0 | 1 | 9 |
| Ill appearing | 14 | 2 | 11 | 27 |
| **Total** | 125 | 8 | 26 | 159 |

**Table 5** Intra-rater agreement between initial gestalt assessment and assessment following examination of overall clinical appearance for the first and second raters.

| First rater gestalt impression | First rater after examining child | | | Total |
|---|---|---|---|---|
| | Not ill appearing | Unsure | Ill appearing | |
| Not ill appearing | 113 | 3 | 2 | 118 |
| Unsure | 8 | 4 | 2 | 14 |
| Ill appearing | 2 | 2 | 23 | 27 |
| Total | 123 | 9 | 27 | 159 |

| Second rater gestalt impression | Second rater after examining child | | | Total |
|---|---|---|---|---|
| | Not ill appearing | Unsure | Ill appearing | |
| Not ill appearing | 113 | 3 | 4 | 120 |
| Unsure | 9 | 5 | 2 | 16 |
| Ill appearing | 3 | 0 | 20 | 23 |
| Total | 125 | 8 | 26 | 159 |

When doctors differ in their 'gestalt' evaluation of a febrile child's overall appearance both of them doing a detailed examination of the patient will narrow their differences.

The frequency with which providers of different training levels chose each classification is shown in Fig. 2. None of our analyses demonstrated a significant effect for level of training and agreement. Inter and intra-rater agreement is shown in Fig. 3. Inter-rater agreement improved with examination compared to gestalt assessment. Table 6 (further expanded in Appendix S3) provides various $\kappa$, $\pi$, polychoric and $AC_1$ and $AC_2$ statistics

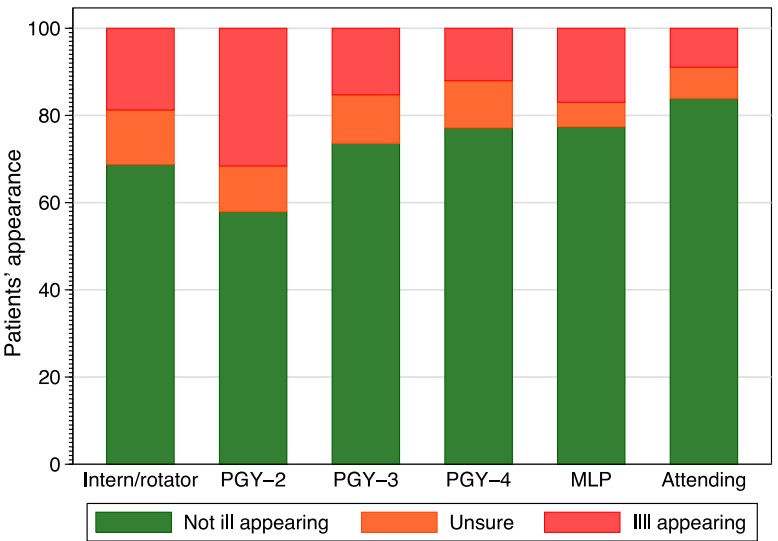

**Figure 2 Classification selected and provider training.** Frequency of classification selected by provider experience. PGY, post graduate year, MLP, mid-level provider.

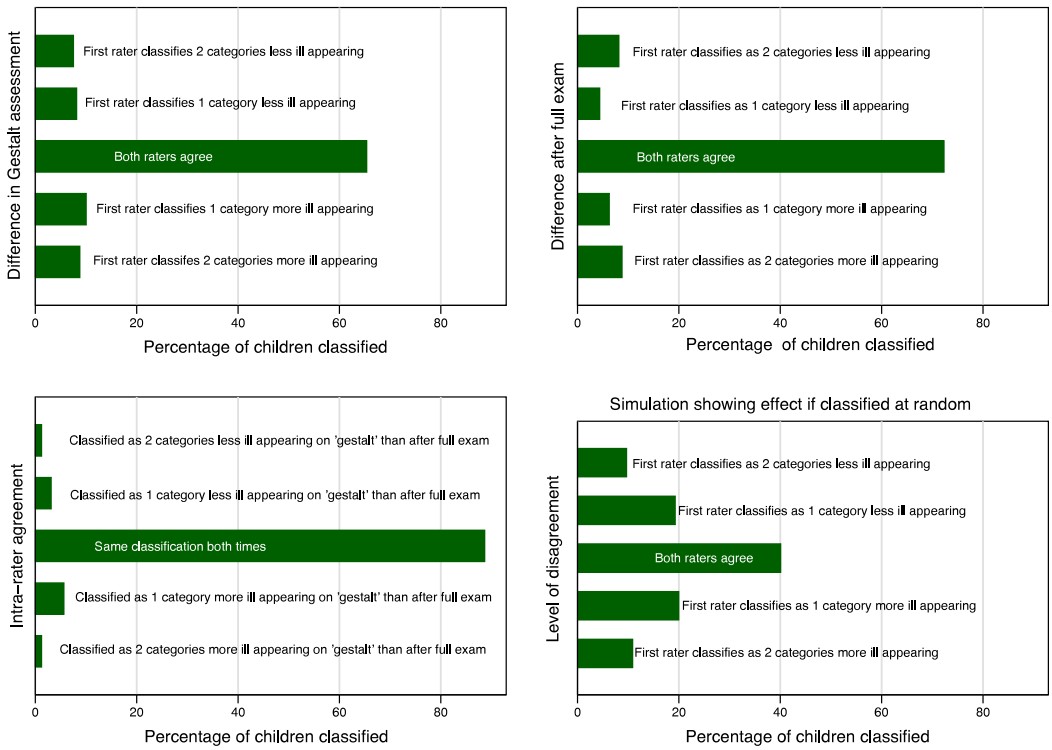

**Figure 3 Graphical analysis of agreement between examiners.** Agreement between examiners' initial 'gestalt' impression, agreement between examiners' after completing their exam, and a simulation showing a uniform random agreement.

**Table 6 Inter rater reliability summary measures.** Inter rater reliability measured by Cohen's $\kappa$, weighted $\kappa$ using two commonly employed weighting schemes, polychoric correlation, and Gwet's AC1.

| | Cohen's $\kappa$ | Weighted $\kappa$[a] | Weighted $\kappa$[b] | Scott's $\pi$ | Scott's $\pi(l)$ | Scott's $\pi(q)$ | Polychoric correlation | Gwet's AC1 | Gwet's AC2($I$) | Gwet's AC2($q$) |
|---|---|---|---|---|---|---|---|---|---|---|
| r-rater Gestalt | 0.119 | 0.181 | 0.223 | 0.118 | 0.177 | 0.261 | 0.334 | 0.550 | 0.601 | 0.635 |
| Inter-rater After exam | 0.235 | 0.283 | 0.314 | 0.216 | 0.261 | 0.289 | 0.482 | 0.655 | 0.672 | 0.683 |
| Intra-rater First rater | 0.690 | 0.781 | 0.844 | 0.695 | 0.777 | 0.833 | 0.955 | 0.852 | 0.893 | 0.920 |
| Intra-rater Second rater | 0.651 | 0.714 | 0.758 | 0.671 | 0.734 | 0.777 | 0.912 | 0.837 | 0.871 | 0.893 |

**Notes.**

[a] Weights $1 - |i - j|/(k - 1)$.

[b] Weights $1 - [(i - j)/(k - 1)]^2$ where $i$ and $j$ index the rows and columns of the ratings by the raters and $k$ is the maximum number of possible ratings, ($l$) linear weighted, ($q$) quadratic weighted. This table is expanded to include other measures of inter-rater agreement in the appendices.

for the results in Table 6. All of these point to increasing agreement when more clinical information is obtained. This also suggests a practical solution for clinicians when faced with uncertainty, either go back and examine the child again or ask a colleague to do so.

There was some asymmetry in the graphs portraying intra rater reliability (Fig. 3). This suggests that a full exam may lead a provider to revise their impression toward increasing the severity of the child's appearance of illness more often than revising their impression towards decreasing the severity of the child's appearance of illness.

There was also slight asymmetry in the graphs describing inter rater reliability favoring increase in the overall appearance of illness by the second reviewer. Sensitivity analysis (excluding 6 outlier cases where the interval between the raters' evaluations exceed 4 h) showed that age (months), odds ratio (OR) 0.89, (95% CI 0.84, 0.95) and time between evaluations (10 min intervals), OR 0.89 (95% CI 0.81, 0.99) impacted inter-rater agreement.

## Sensitivity analysis

Antipyretic use before or between evaluations (even when interacted with the time interval between evaluations and age), experience of the provider pair, diagnosis, and infant age less than 2 months, were all non-significant. The results were similar when the outlier cases were included. The model fit and calibration were satisfactory (with and without the outlying cases) and are detailed in Appendix S5. The inter-rater agreement of overall clinical impression between providers was greatest when the child was considered 'not ill appearing'.

We found a wide range of values that could be calculated for different $\kappa$ variants and other measures of agreement and the very low values of traditional marginal agreement statistics. Many of these could reasonably be presented as a true reflection of the inter rater-reliability of provider's assessment of a febrile child's overall appearance. Based on most of the inter-rater reliability measures; overall clinical appearance, the central tenet of management algorithms for febrile infants and toddlers would be discarded as unreliable. However our graphical analysis portrays a different picture entirely. This picture is one of overwhelming agreement between clinicians, with the caution that a second or closer look may find evidence of increasingly ill appearance. Of the summary statistics of agreement

only Gwet's AC provided an estimate that would allow a reader to intuit the agreement observed given the imbalance between ill and not ill appearing children.

## DISCUSSION

The inter-rater reliability of ED provider assessment of overall clinical appearance in febrile children aged less than 24 months was modest. Inter-rater agreement was better after providers had examined the child than when they relied solely on gestalt. Agreement decreased in older children, and as the time interval between the two providers' evaluations increased. Classifications of 'not ill appearing' were more reliable than others. Provider experience level had little effect on agreement.

Different summary measures of agreement, and different weighting schemes yielded different results. Graphical portrayal of these results better communicated the inter-rater reliability than did the single summary statistical measures of agreement. Among the summary statistical measures of agreement Gwet's AC most closely paralleled the graphical presentation of results.

Our results are broadly consistent with those of *Wagai et al. (2009)* that compared clinicians' evaluations using videos of children. We have previously used videos for training and measuring inter-rater reliability in the Face, Legs, Activity, Cry and Consolability (FLACC) pain score in infants. Videos allow standardization of inter-rater reliability measurements. The disadvantage of videos, however, is a loss of validity as an artificial situation is created. Our finding that clinical experience did not affect agreement of overall clinical appearance is consistent with the finding of Van der Bruel that found that the seniority of the physician did not affect the diagnostic importance of a 'gut feeling that something is wrong' in 3,981 patients aged 0–16 years (*Van den Bruel et al., 2012*). Our findings differ from prior work in children aged less than 18 months with bronchiolitis (*Walsh et al., 2006*). This may be because of inherent differences in the conditions.

The use of the $\kappa$ statistic is often an appropriate strategy for analyzing studies of inter-rater reliability. However the apparent paradox where high actual agreement can be associated with a low or even negative $\kappa$ can mislead rather than enlighten. This was clearly evident in our study, the weighted $\kappa$ was 0.223. Experiments where examiners were told to guess their physical findings based on a third party clinical report have been used to argue that low $\kappa$ statistics in fact reflect the true reliability of examination findings in children (*Gorelick & Yen, 2006*). The difficulty for clinicians accepting such a strategy is that it appears to lack face validity. Clinical outcomes do not demonstrate the variability in outcome that one would intuitively expect were clinical examination so unreliable.

We have also shown how differently weighted $\kappa$ and other measures of agreement may differ substantially from each other (*Gwet, 2012*). Some have recommended focusing on those cases in which disagreement might be anticipated (*Lantz & Nebenzahl, 1996*; *Vach, 2005*); others recommend abandoning $\kappa$ (or the proposed experiment) entirely where expected prevalence is not ∼50% (*Hoehler, 2000*). A compromise approach has been to argue that in addition to the omnibus $\kappa$, percentages of positive and negative agreement should also be presented (*Cicchetti & Feinstein, 1990*).

This compromise approach is not dissimilar to our graphical solution; a simple graph is readily grasped and allows the reader detect asymmetry which may suggest changes between the ratings. Graphs have the attraction of allowing relative complex data to be absorbed quickly by readers, even if the reader has little or no statistical training.

Graphical methods also have disadvantages. Graphs require more space than a single summary statistic and are more difficult to summarize. A number is easier to communicate verbally to a colleague than a graph. Different readers may view the same graph and disagree about its meaning raising the question by whose eye should a graph be judged? Another limitation of graphical methods is their vulnerability to axis manipulation or failure to include a reference graph of what agreement by random chance alone would look like using the same scale for each axis. Portraying simple differences in agreement graphically may not however be the optimal solution for every situation.

The apparent objectivity and simplicity of a single number makes decision making easier. However we argue that summary statistics also increase the risk of the wrong decision being made as to whether or not a characteristic is sufficiently reliable be included in decision making. When graphical methods are not optimal, providing separate summaries of the proportionate agreement in each class, or Gwet's $AC_{1\&2}$ may be an alternative. Certainly, it seems unwise to discount clinical findings for inclusion in prediction rules and management algorithms solely based on $\kappa$ scores of $<0.5$, without consideration of other measures. This is particularly the case where categories are expected to be highly unbalanced as in, for example, serious bacterial infection in infants with bronchiolitis, intracranial bleeding in head injury and cervical spine injury in blunt trauma (*Chee et al., 2010*; *Leonard et al., 2011*; *Kuppermann et al., 2009*).

## Limitations

There are several limitations to our work. Data were collected at a single teaching hospital and the results may not be generalizable to other sites. Although a diverse mix of providers was measured, the lack of attending physician to attending physician comparisons decreases generalizability. We also assumed that within categories of raters are interchangeable. This unproven assumption is typical of research evaluating the reliability of clinical signs for inclusion in diagnostic or treatment algorithms. Few very sick infants were included. Similarly for the 'unsure' category our number of patients was very small. This may be because of the rarity of conditions such as bacterial meningitis and because of physicians' unwillingness to enroll very sick infants in the study out of concern that study enrollment would delay care or disposition. Such concerns have hampered previous attempts at measuring the inter rater reliability of the Ottawa ankle rule in children and may help explain the dearth of inter rater studies in the evaluation of febrile infants (*Plint et al., 1999*).

Our sample size is small. This is partly because during the pilot phase it emerged that data integrity would require a careful policing by research assistants to ensure the providers would not discuss cases and jeopardize the independence of their assessments. As a result,

when only a single RA was on duty it is possible some potential cases were missed if more than one febrile infant or toddler presented simultaneously.

Our design had many different pairs of raters evaluate many different infants. Although a widely used approach (*Holmes et al., 2005*; *Kuppermann et al., 2009*; *Plint et al., 1999*; *Walsh et al., 2006*), this design introduced additional variance to the agreement coefficients than would occur in a study where all the raters examined the same infants. This additional variance decreases the generalizability of our findings. Avoiding this increased variance would, as a practical matter, require that investigators record videos of febrile children and have all raters review the same videos. Such designs, however, introduce additional sources of bias; namely those of the cameraman and video editor. Moreover we felt that important aspects of the clinical encounter used in judging infant and toddler appearance are also lost in video recordings.

## CONCLUSION

The inter-rater reliability of EP assessment of overall clinical appearance was adequate. Inter-rater reliability is sometimes better described graphically than by a summary statistic; different summary statistics yield different results for the same dataset. Inter-rater agreement of overall appearance should not always be reduced to a single summary statistic but when categories are unbalanced Gwet's AC is preferred.

### Funding

This work was supported by The Pediatric Emergency Medicine Research Foundation, Long Beach, CA and by Award Number 5K12HL108964-02 from the National Heart, Lung, and Blood Institute at the National Institutes for Health, the National Center for Advancing Translational Sciences, National Institutes of Health, through grant number UL1 TR000002. The content is solely the responsibility of the authors and does not necessarily represent the official views of the National Heart, Lung, and Blood Institute or the National Institutes of Health or The Pediatric Emergency Medicine Research Foundation. The funders had no role in study design, data collection and analysis, decision to publish, or preparation of the manuscript.

### Grant Disclosures

The following grant information was disclosed by the authors:
The Pediatric Emergency Medicine Research Foundation.
National Heart, Lung, and Blood Institute: 5K12HL108964-02.
National Center for Advancing Translational Sciences, National Institutes of Health: UL1 TR000002.

### Competing Interests

The author declare there are no competing interests.

## Author Contributions

- Paul Walsh conceived and designed the experiments, performed the experiments, analyzed the data, contributed reagents/materials/analysis tools, wrote the paper, prepared figures and/or tables, reviewed drafts of the paper.
- Justin Thornton conceived and designed the experiments, performed the experiments, reviewed drafts of the paper.
- Julie Asato, Nicholas Walker, Gary McCoy, Joe Baal, Jed Baal, Nanse Mendoza and Faried Banimahd performed the experiments, reviewed drafts of the paper.

## Human Ethics

The following information was supplied relating to ethical approvals (i.e., approving body and any reference numbers):

Kern Medical Center IRB approval #10032.

## Supplemental Information

Supplemental information for this article can be found online at http://dx.doi.org/10.7717/peerj.651#supplemental-information.

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
