# Peer review of "Approaches to describing inter-rater reliability of the overall clinical appearance of febrile infants and toddlers in the emergency department"

_PeerJ, doi:10.7717/peerj.651_

## Round 0.1 · original submission · Major Revisions

· Academic Editor

Major Revisions

I concur with the reviewers that a more careful description of the clinical study is needed. The reviewers also identify several important limitations to the study and these should be explicitly acknowledged and the potential consequences for the validity of the conclusions should be discussed.

·

Basic reporting

Overall, I found this article very well written. The authors motivate their research in a clear manner in the introductory section, and the flow of ideas is easy to follow. By and large this paper is interesting and reads well. Although the focus was on the inter-rater reliability among clinicians about the clinical appearance of children, the paper has also mentioned intra-rater agreement (Tables 4A and 4B). However, intra-rater reliability is not discussed much in this paper, making me wonder why it was reported in the first place.

Figure 1 is very trivial, and most readers would probably not miss it if it was removed. I may be less familiar with the sunflower plot than other researchers. But I did struggle a little to grasp the information that is conveyed by Figure 2. Perhaps a little more explanation will help here.

Experimental design

The authors have defined the research question very well. They wanted to quantify the extent of agreement among clinicians about the clinical appearance of children with fever aged 24 months or younger. This investigation is certainly essential for clinicians working in a similar environment. The use of the different agreement coefficients, including kappa, AC1/AC2 is adequate.

The design of this experiment has some problems that the authors must acknowledge in their paper. The main problem is the arbitrary way in which clinicians are assigned to the children being examined, and the fact that different pairs of clinicians rated different children. The authors have discussed this in terms of bias, and have done a simulation to demonstrate that this bias is minimal. Doing this sensitivity analysis shows that the authors have taken this problem seriously. However, the bias is not the real problem here. The real problem is the increased variance of the agreement coefficients due to the random assignment of clinicians to the patients. Normally, when dealing with multiple raters, the correct solution is often to have each rater rate all patients. This avoids adding an extra source of variation in the magnitude of the agreement coefficient. Now, I know very well that this optimal design is often impractical in most clinical settings. Few parents will allow their infants to be examined by multiple clinicians for the sole purpose of quantifying inter-rater reliability. That is why the use of videos is often helpful in this context. Agreement coefficients that are subject to large variances are also more difficult to generalize. The authors could simply mention these limitations in the revised manuscript.

Validity of the findings

The authors have realized like many others before that when evaluation inter-rater reliability in the context of high or low trait prevalence, the kappa coefficient may yield unpredictable results, and that alternative measures were needed. But they illustrate this issue well in the context of medical triage of children with fever in an emergency department. The data tables are informative and well presented.

The last discussion section describes the study limitations very well. However, the authors should add some of the issues raised in the "Experimental Design" section of this review.

Additional comments

No Comments.

·

Basic reporting

The paper now hinges on being a methodological paper and a clinical paper. There is only limited information on the methods of the data collection, and very lengthy information on the statistics. The paper could be written more clearly by purely focusing on the methodological side of things, and reporting the clinical results separately.
Tables 2-4 are difficult to interpret and should be revised.
The box in the main body of the manuscript inserted between lines 190-191 contains an error: the last row should read "less ill".

Experimental design

There is insufficient information on the data collection: how were providers selected? Were they asked to include every eligible child (consecutive inclusion)? What was the maximum time allowed between the two assessments?
How long did the data collection last, as each provider included on average 5 children? Were there any imbalances in the number of children included per provider?
What information was provided to the participating clinicians on how to rate the children? In particular, this is important for the unsure category: was it defined as neither ill nor unwell appearing, or impossible to draw a conclusion because of lack of cooperation etc?

Validity of the findings

There is insufficient information on the time between the two assessments. On average, how much time was there between two assessments, and what was the minimum and maximum? This is vital as there may be a number of reasons why there might be a change in the clinical presentation: the child may have actually improved/deteriorated, there may be an effect of antipyretics provided in between assessments, there may be regression to the mean.
Table 5 includes intra-rater results: what does this refer to? Have raters been asked to rate the same child twice?
What level of significance was used in the sensitivity analyses? How many interactions were taken into account? Were the variables selected backwards or forwards? What was the goodness of fit and calibration of the final model?

---

## Round 0.2 · accepted · Accept

· Academic Editor

Accept

The revised version of the manuscript has adequately addressed the concerns raised in the review process.